# Comfort or conservation? Investigating patient choices between plastic and metal speculums

**Torren A. Kalaskey**[1]*, **Kathryn G. Quillen**[2], **Kyle R. Knutsen**[1], **Kylie R. Bushko**[1], **Omar F. Dueñas-Garcia**[2]

**1** WVU School of Medicine, 64 Medical Center Dr, Morgantown, West Virginia, United States of America, **2** Department of Obstetrics and Gynecology, WVU Medicine, 1 Medical Center Dr, Morgantown, West Virginia, United States of America

* tak00011@mix.wvu.edu

## Abstract

### Background

The pelvic examination is essential in gynecologic care but often causes discomfort and anxiety. Choice of plastic or metal speculums may influence patient comfort and adherence to screening, yet patient preferences and environmental considerations remain underexplored.

### Objective

To assess patient preferences between plastic and metal speculums, identify factors influencing these preferences, and evaluate environmental concerns related to speculum use.

### Methods

A cross-sectional survey was conducted among 203 patients receiving care at West Virginia University's Obstetrics and Gynecology clinics between September 15, 2024 and June 16, 2025. Participants completed an anonymous online questionnaire assessing speculum preference, comfort factors, sanitation perceptions, and environmental awareness. Data were analyzed using descriptive statistics and chi-square tests.

### Results

Plastic speculums were preferred by 49.8% of respondents, followed by no preference (32.5%) and metal speculums (17.7%). Younger participants (ages 18–35) showed a stronger preference for plastic. The speculum was identified as the most uncomfortable aspect of the exam by 34.4% of respondents, with temperature and positioning also frequently cited. Plastic was perceived as more sanitary by 45.8%. Environmental concern about plastic waste was higher among younger respondents

**Data availability statement:** All data files are publicaly available from the Open Science Framework (OSF) repository. URL: https://osf.io/5dxte/overview DOI: 10.17605/OSF.IO/5DXTE.

**Funding:** The author(s) received no specific funding for this work.

**Competing interests:** The authors have declared that no competing interests exist.

and those preferring metal speculums. Despite ecological awareness, comfort during the exam was the predominant factor influencing preference.

## Conclusions

Patient discomfort remains a primary barrier to maintaining consistent patient care in the field of obstetrics and gynecology, and this research found that the majority favor plastic speculums due to comfort despite environmental concerns in their lack of reusability. Efforts to improve pelvic exam experiences should address both comfort and sustainability, including innovations in speculum design, patient education on reusable options, and alternative screening methods. Further research is warranted to balance patient-centered care with environmental stewardship.

### Introduction

The pelvic examination is a cornerstone of obstetrics and gynecology (OB/GYN) care, essential for diagnosing gynecologic conditions, preventing and detecting cervical cancer, collecting specimens, and promoting reproductive and sexual health [1]. The exam typically includes inspection of the external genitalia, a speculum exam to visualize the vagina and cervix, and a bimanual examination to assess the uterus and adnexa. Despite its clinical importance, the pelvic exam can be a significant source of anxiety and discomfort for patients due to physical discomfort and the inherent vulnerability of the experience [1–3]. This distress may be further heightened in individuals with a history of sexual trauma [1,4]. Numerous factors influence patient comfort during the exam, including the quality of the patient-provider interaction and the equipment used—most notably, the speculum itself [5]. Speculums are available in a range of sizes and are typically made from either plastic or metal [6]. Understanding patient preferences regarding speculum material improves patient comfort and adherence to guideline-recommended pelvic exams.

The United States leads the world in the use of single-use disposable plastic medical supplies, generating an estimated 1.7 million tons of plastic waste annually [7]. Overall, the U.S. healthcare system is responsible for approximately 8.5% of the nation's total greenhouse gas emissions [7]. Among these supplies, disposable plastic speculums contribute substantially to medical plastic waste [8,9]. In 2023, an estimated 420 million disposable plastic speculums were used globally, with North America accounting for 35% of consumption and Asia 30% [7]. Cost of acquisition, operation, sterilization, and disposal of metal speculums are higher over a 5 year period than cost of acquisition and disposal of plastic speculums [9]. Additionally, a study found the majority of physicians prefer using plastic over metal speculums, both for handling purposes related to the pelvic exam and patient comfort, suggesting there may be an increase in plastic use [9].

Despite the widespread use of both metal and plastic speculums, limited research has explored patient preferences between the two. Metal speculums, introduced in the 1800s, remained the standard until the 1960s, when plastic speculums were

developed as a disposable alternative [10]. As global awareness of cervical cancer screening and access to gynecologic care increase, particularly in historically underserved regions, overall speculum is expected to increase [11]. Additionally, the growing use of disposable medical equipment in healthcare systems, driven in part by reduced time and costs, may further increase the use of plastic speculums [8]. This trend has significant environmental implications, including increased greenhouse gas emissions and plastic pollution [8,12]. From an environmental perspective, a shift back to reusable metal speculums is preferable due to reduced waste and carbon impact [7,8,12,13].

However, patient preference remains a critical factor in ensuring a positive examination experience, as discomfort can discourage future screening adherence [2]. A cross-sectional study conducted across five Dutch hospitals found that while most patients preferred plastic speculums, many were open to compromise in support of a more environmentally sustainable healthcare system [14].

To further investigate these preferences, a cross-sectional survey was conducted among patients receiving care through the West Virginia University Department of Obstetrics and Gynecology. The study aimed to determine whether patients had a preference between metal and plastic speculums and to identify the factors driving those preferences.

## Methods

### Patient and public involvement

This study was motivated by the well-documented discomfort and apprehension many patients experience during pelvic examinations, particularly speculum use. The primary objective was to assess patient preference between plastic and metal speculums. Secondary objectives included exploring the reasons behind those preferences, with the aim of informing future interventions to improve patient comfort and reduce barriers to routine gynecologic care.

### Study population and recruitment

Participants were recruited from outpatient clinics affiliated with the West Virginia University (WVU) Department of Obstetrics and Gynecology. Eligible patients were invited to participate during routine clinic visits and were provided with a QR code linking directly to the online survey. Upon scanning the code using their personal mobile devices, participants were presented with an electronic consent form. Participation was entirely voluntary and anonymous; completion of the survey served as implied informed consent.

### Study design and sampling

This study utilized a convenience sample of consecutive patients presenting to outpatient obstetrics and gynecology clinics during the recruitment period. No a priori power calculation was performed, as the study was designed as an exploratory assessment of patient preferences.

### Survey instrument

The survey instrument was developed by the research team based on literature review and clinical expertise. Questions were pilot-tested among a small group of trainees and faculty for clarity and content validity prior to distribution. No previously validated instrument addressing speculum material preference was identified.

### Response rate

All eligible patients present during clinic sessions were invited to participate until the pre-decided goal of roughly 200 surveys was obtained. The number of patients who declined participation was not systematically recorded; therefore, an exact response rate cannot be calculated. This may introduce potential selection bias.

## Ethical approval and data collection

The study protocol was reviewed and approved by the WVU Institutional Review Board (IRB). Data collection was conducted over an eight-month period, from September 15th, 2024 through June 16th, 2025. Survey responses were securely collected and stored using the Qualtrics survey platform. Upon survey completion, participants were given the option to view a summary of aggregate results via the same QR code link.

## Data analysis

Descriptive statistics were used to summarize participant demographics and speculum preferences. No identifying individual information was accessed during data collection or analysis. Data was accessed on June 21, 2025 for analysis. Associations between categorical variables were evaluated using chi-square tests. Given the exploratory nature of secondary analyses, no formal correction for multiple comparisons was applied; findings should therefore be interpreted as hypothesis-generating.

Analyses were conducted using complete-case analysis for each variable, excluding missing responses from individual comparisons while retaining overall sample counts. Percentages may not total 100% due to rounding.

Statistical analyses were conducted using Stata 13 (StataCorp LLC, College Station, TX). Additionally, OpenAI's ChatGPT (GPT-4, 2025) was used as a supplemental tool for automated data processing, visualization development, and exploratory insight generation. The SRQR reporting guideline was used to draft this manuscript, and the SRQR reporting checklist was used when editing.

## Results

A total of 207 survey records were exported from Qualtrics. Four survey preview entries were excluded, resulting in 203 non-preview responses included in the analysis. Of these, 197 respondents completed the survey in full, while 6 provided partial responses. Item-level missingness resulted in varying denominators across analyses. For the primary outcome of speculum preference, 199 respondents provided non-missing responses. The largest proportion of respondents were between 25–35 years old (n = 78), followed by 18–24 years (n = 47), 35–49 years (n = 41), 50–65 years (n = 29), and over 65 years (n = 8).

As shown in Fig 1, plastic speculums were the most preferred overall (n = 101), followed by respondents who had no preference (n = 66), and those who preferred metal speculums (n = 36). Fig 2 illustrates preference

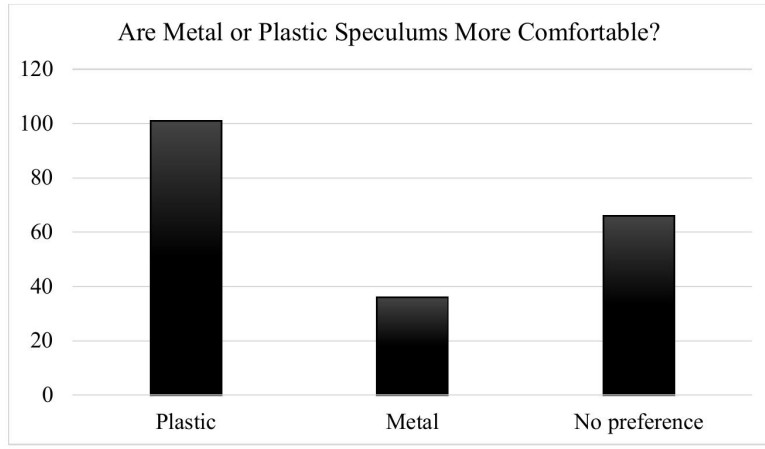

**Fig 1. Total Speculum Comfort Preferences.**

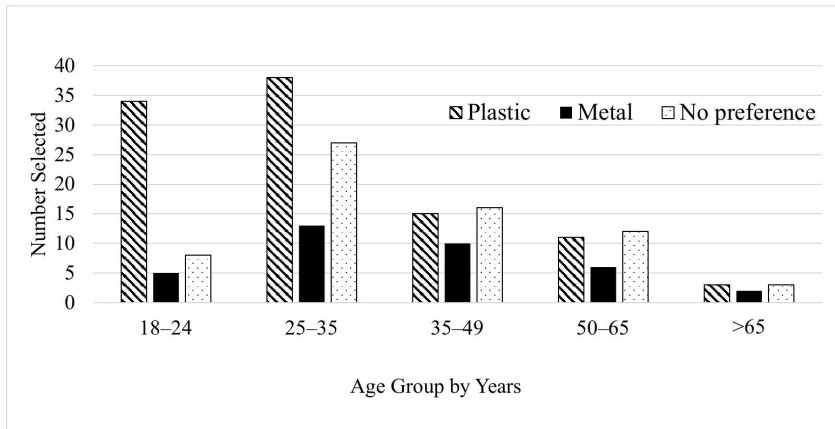

**Fig 2. Speculum Preference By Age Group.**

distribution by age group, with younger participants (ages 18–35) demonstrating the strongest preference for plastic. A chi-square test demonstrated a borderline association between age group and speculum preference ($\chi^2(8) = 14.36$, $p = 0.073$).

To better understand the factors contributing to these preferences, participants were asked which type of speculum they believed to be more sanitary. Plastic was perceived as more sanitary by 93 participants (45.8%), while 38 participants (18.7%) selected metal, and 68 (33.5%) indicated both were equally sanitary.

Participants were also asked to identify the most uncomfortable aspect of the speculum exam; results are presented in Table 1. Those who preferred plastic speculums were prompted to select the primary reason for their preference. The most cited reasons included more comfortable insertion (n = 42), better temperature (n = 29), and perceived sanitation (n = 16). Among those who preferred metal speculums, top reasons included more comfortable insertion (n = 13), perceived sanitation (n = 8), and lower environmental impact (n = 8).

Environmental concern regarding plastic waste was more prevalent among younger respondents. When asked, "Considering plastic speculums are not reusable, do you have any concern about the carbon footprint and waste that it generates?", 28 participants who preferred metal speculums answered yes, compared to 23 among those who preferred plastic. Notably, environmental concern was most common among respondents aged 25–35 (n = 35) and 35–49 (n = 24), whereas concern was lowest among those older than 65.

**Table 1. What is the most uncomfortable part of a pelvic exam?.**

| What is the most uncomfortable part of a pelvic exam? | Responses Count | Percent of Total* |
|---|---|---|
| Unable to See Exam | 18 | 9.2% |
| Temperature of Speculum | 31 | 15.9% |
| The Speculum | 67 | 34.4% |
| Position of Exam | 35 | 17.9% |
| Lack of lubrication | 18 | 9.2% |
| Not Unpleasant | 26 | 13.3% |

*Percentages may not total 100% due to rounding.

## Discussion

This study investigated patient preferences between plastic and metal speculums and examined how those preferences relate to comfort, perception of sanitation, and environmental concerns. Although the study found that plastic speculums were preferred most of all options, this preference highlights the ongoing tension between prioritizing patient comfort and advancing environmentally sustainable healthcare practices.

### Study limitations

Several limitations must be acknowledged. Because this study employed a convenience sample without an a priori power calculation, it may have been underpowered to detect small subgroup differences, including age-related associations (disproportionate number of respondents being in the younger age groups). Additionally, the demographic diversity of the sample was limited; the majority of participants identified as cisgender women, making it difficult to draw conclusions about speculum preference across gender identities. As such, the findings may not be generalizable to more diverse or gender-diverse populations. Lastly, the survey did not collect detailed demographic variables such as race, ethnicity, parity, socioeconomic status, or rural versus urban residence. West Virginia has a predominantly non-Hispanic White population (~92%), which may limit generalizability to more diverse populations [15].

### Clinical implications and opportunities for improvement

The results highlight the importance of prioritizing patient comfort during pelvic exams. Although plastic speculums were perceived as more comfortable, particularly with regard to insertion and temperature, the speculum itself was not the sole source of discomfort. In this study, only 34% of participants reported the speculum as the most uncomfortable aspect of the exam, while temperature and positioning were also frequently cited (Table 1). These findings suggest that efforts to improve the pelvic exam experience should extend beyond speculum design alone.

One avenue for improvement involves innovation in speculum design. For instance, the Bouquet Speculum, a five-petaled model designed to distribute pressure evenly, has shown promise in improving comfort and visualization [6]. Further studies are warranted to evaluate its impact on patient experience and exam outcomes.

Temperature sensitivity was also a common concern, particularly among those preferring plastic. While warming systems like those used for ultrasound gel may be considered, evidence supporting their effectiveness is limited. One emergency department study found no significant improvement in patient satisfaction when warmed gel was used during point-of-care ultrasound [5]. Further research is needed before recommending widespread adoption of warming systems for speculums.

Positioning is another factor worth revisiting. A clinical trial found that patients reported decreased vulnerability and discomfort when their feet were placed on the edge of the exam table rather than in stirrups, without compromising the quality of Pap smears [3]. This simple modification may be an effective strategy to improve patient experience.

### Environmental trade-offs and global perspectives

This study's findings align with a previous cross-sectional study conducted in five Dutch hospitals, which also found that the majority of patients preferred plastic speculums [14]. However, a key difference was that participants in the Dutch study were more willing to compromise personal comfort for ecological sustainability, while most participants in our study expressed minimal concern about the environmental impact of plastic use as displayed in Table 2 [8,12,13]. However, survey-based studies are more susceptible to social desirability bias, which may influence results. Participants may more likely to report environmental-conscious preference as it is a socially desirable attitude, leading to an overestimation of pro-environmental attitudes [16]. The extend of this bias may vary across populations and cultures contexts, and thus these findings my not fully reflect patient preferences in real-world clinical settings when choosing between metal and plastic speculums [16].

**Table 2. Carbon Footprint Comparison: Plastic vs. Metal Vaginal Speculums [7,8,13].**

| Criteria | Disposable Plastic Speculum | Reusable Metal Speculum (e.g., stainless steel) | Primary Source |
|---|---|---|---|
| Material Type | Polystyrene, Polypropylene, or other medical-grade plastic | Stainless steel (typically 304 or 316 grade) | 7,8 |
| Use Type | Single-use only | Reusable (up to 500 times) | 7,8,13 |
| Carbon Footprint per 20 Speculum Exams | ~17.54 kg $CO_2$ equivalents | ~5.72–6.51 kg $CO_2$ equivalents per use (after accounting for reuse & sterilization) | 8 |
| Carbon Footprint per 500 Speculum Exams | ~438.55 kg CO2 equivalents | ~101.31–107.52 kg CO2 equivalents | 8 |
| Sterilization Required | No | Yes (autoclaving or chemical sterilization) | 8,13 |
| Sterilization $CO_2$ Contribution | N/A | ~4.58–12.53 kg $CO_2$e per cycle (depends on energy source, capacity loaded, and method) | 8 |
| Disposal Impact | High (landfill or incineration) | Low (after lifetime, recyclable metal) | 8 |
| Lifespan | One patient use | Up to 500 uses | 8 |
| Recyclability | Poor (contaminated plastic, often not recyclable) | High (metal recyclable after final use) | 8 |
| Overall Environmental Impact | High (due to single-use and plastic waste) | Lower (despite energy for sterilization) | 8 |

Despite the preference for plastic speculums, younger respondents (ages 18–35) were more likely to express environmental concern than older respondents. This discrepancy may indicate that although ecological awareness exists among younger individuals, immediate comfort during an exam could influence preference of speculum more strongly than longer-term environmental considerations. The survey did not directly assess trade-off decision-making between comfort and sustainability; therefore, interpretations regarding prioritization should be considered exploratory. Future research using a discrete choice experiment design would quantify the relative importance of specific attributes of the pelvic exam, such as patient positioning, temperature of speculum, type of speculum, and other contributing factors [17]. Additionally, a randomized crossover study in which patients rate pain, comfort, and overall preference following exposure to both metal and plastic speculums in randomized order would provide prospective, within-subject comparisons and reduce reliance on recall bias present in survey-based designs.

The carbon footprint created from using plastic speculums stems from production, transportation, and waste management of a non-recyclable material [8]. Comparatively, the carbon footprint related to metal speculum use is due to sterilization, either through autoclaving or H2O2 (7.25%), and reduction in emissions from autoclaving can be made by striving to sterilize with full capacity compared to only half full [8]. Reusable metal speculums offer a more sustainable alternative overall after as few as 10 speculum exams [8]. These instruments can be reused up to 500 times, and although the sterilization process contributes to carbon emissions, the overall environmental impact is lower compared to manufacturing single-use plastic devices [7,8,12,13]. However, despite adequate sterile protocols for both metal and plastic speculum handling, metal speculums were perceived as less sanitary in this study, suggesting that patient education on the sterilization process of metal speculums may help increase acceptance of reusable options.

In addition to design modifications, alternative screening approaches such as self-collected vaginal swabs for HPV testing may reduce the need for speculum exams altogether. These have been shown to be as acceptable as clinician-collected samples for primary HPV screening while significantly reducing material use [18]. A study from the UK reported an 8.7-fold reduction in environmental impact using self-sampling [19]. The self-collection method could be particularly beneficial for patients who avoid pelvic exams due to discomfort or trauma histories.

## Conclusions

This study found that plastic was most selected type of speculum material due to comfort of the pelvic exam and other factors. Through surveying 203 patients, several potential areas for improvement in the pelvic examination process were identified: increasing patient comfort, promoting sustainable speculum use, and adopting innovative screening alternatives. Given the discrepancy between patient preference and environmental sustainability, long-term solutions should include not only enhanced exam techniques and improved speculum design but also increased patient education and availability of eco-friendly options. Additional research is needed to evaluate the importance of specific attributes of the pelvic exam as well as the feasibility and effectiveness of implementing changes in clinical practice.

## Supporting information

**S1 File. SRQR reporting checklist.**
(DOCX)

## Acknowledgments

The authors would like to thank the Department of Obstetrics and Gynecology at West Virginia University Medicine for supporting this project and providing access to data.

## Author contributions

**Conceptualization:** Kyle R Knutsen, Omar F Dueñas-Garcia.

**Data curation:** Torren Alexa Kalaskey, Kyle R Knutsen, Kylie R Bushko.

**Formal analysis:** Kathryn G Quillen, Omar F Dueñas-Garcia.

**Investigation:** Torren Alexa Kalaskey, Omar F Dueñas-Garcia.

**Methodology:** Omar F Dueñas-Garcia.

**Project administration:** Omar F Dueñas-Garcia.

**Supervision:** Kathryn G Quillen, Omar F Dueñas-Garcia.

**Validation:** Kathryn G Quillen, Omar F Dueñas-Garcia.

**Visualization:** Torren Alexa Kalaskey, Omar F Dueñas-Garcia.

**Writing – original draft:** Torren Alexa Kalaskey, Kyle R Knutsen.

**Writing – review & editing:** Torren Alexa Kalaskey, Kathryn G Quillen, Kyle R Knutsen, Kylie R Bushko, Omar F Dueñas-Garcia.

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
