## [Decision Letter · Decision Letter 0]

11 Feb 2026

PONE-D-25-65007Comfort or Conservation? Investigating Patient Choices Between Plastic and Metal SpeculumsPLOS One?

Dear Dr. Kalaskey,

Thank you for submitting your manuscript to PLOS ONE. After careful consideration, we feel that it has merit but does not fully meet PLOS ONE’s publication criteria as it currently stands. Therefore, we invite you to submit a revised version of the manuscript that addresses the points raised during the review process.

We look forward to receiving your revised manuscript.

Kind regards,

Alessandro Favilli, PhD, MD

Academic Editor

PLOS One

Journal Requirements:

2. In the online submission form, you indicated that “The data is and will continue to be stored in Qualtrics. It can be made available upon request.”

Reviewers' comments:

Reviewer's Responses to Questions

**Comments to the Author**

1. Is the manuscript technically sound, and do the data support the conclusions?

Reviewer #1: Yes

Reviewer #2: Yes

2. Has the statistical analysis been performed appropriately and rigorously?

Reviewer #1: Yes

Reviewer #2: Yes

3. Have the authors made all data underlying the findings in their manuscript fully available?

Reviewer #1: No

Reviewer #2: Yes

4. Is the manuscript presented in an intelligible fashion and written in standard English?

Reviewer #1: Yes

Reviewer #2: Yes

Reviewer #1: The study demonstrates several strengths, including clear research objectives, an appropriate cross sectional design, and an adequate sample size of 203 participants for descriptive analysis. The authors are commended for their transparent disclosure of AI tool usage in data analysis and for including an informative environmental impact comparison table. The acknowledgment of study limitations further strengthens the manuscript’s credibility. However, a few areas warrant clarification or improvement. The data collection period stated in the abstract (November 2024–June 2025) differs from that reported in the Methods section (September 15, 2024–June 16, 2025), and this discrepancy should be addressed. Additionally, providing more demographic details such as race/ethnicity and parity, if collected, would help better characterize the sample. The borderline significant chi-square result for the age preference association would benefit from reporting the actual p-value. Finally, the Data Availability Statement should specify the nature of the restrictions and provide contact information for data requests. No concerns regarding dual publication, research ethics, or competing interests were identified, and the study received appropriate IRB approval with implied consent procedures.

Reviewer #2: The study addresses a clinically relevant and timely question with implications for both patient experience and healthcare sustainability. The research is well-motivated, the methods are appropriate for the research questions, and the discussion thoughtfully contextualizes the findings within broader clinical and environmental considerations.The manuscript addresses an understudied topic that has practical implications for gynecologic practice. The integration of patient comfort preferences with environmental sustainability considerations is novel and timely given increasing attention to healthcare's environmental footprint. The authors provide appropriate ethical oversight documentation and transparent reporting of AI use in data analysis. The discussion section is particularly well-developed, offering practical clinical recommendations including alternative positioning techniques, innovative speculum designs, and self-sampling alternatives for HPV screening. The comparison with the Dutch study adds valuable cross-cultural perspective to the findings.

Minor Issues Requiring Revision

1. Methodological Clarifications

Sample size and power (Lines 161-163), the authors acknowledge the study was underpowered to detect significant associations between age group and speculum preference. A prospective power calculation should be reported, or the authors should clarify whether this was a convenience sample. Additionally, reporting the target sample size and how it was determined would strengthen the methods section.

Recruitment period discrepancy, the abstract states data collection occurred "between November 2024 and June 2025," while the methods section (Lines 102-104) indicates "September 15th, 2024 through June 16th, 2025." The authors should reconcile this inconsistency.

Survey instrument validation, the authors do not describe whether the survey instrument was piloted or validated. Information about how questions were developed, whether they were adapted from existing instruments, and any pilot testing would strengthen confidence in the measurements.

Response rate, the manuscript does not report how many patients were approached versus how many completed the survey. The response rate is important for assessing potential selection bias and should be included.

2. Statistical Analysis

Chi-square results (Line 128-129), the text states "A chi-square test revealed a borderline significant association between age group and speculum preference" but does not report the chi-square statistic, degrees of freedom, or p-value. These statistics should be provided in the text or a supplementary table.

Multiple comparisons, the study appears to conduct multiple statistical tests without correction for multiple comparisons. The authors should address whether any adjustment was made or justify the exploratory nature of the analyses.

Missing data handling, the manuscript does not describe how missing data were handled. Given that the total sample is 203 but some response categories do not sum to this total (e.g., sanitation perceptions in Lines 134-136 sum to 199), clarification is needed.

3. Presentation Issues

Table 1: The percentages in Table 1 sum to 99.9%, which appears to be a rounding artifact but should be verified. Additionally, consider presenting confidence intervals for the percentages to convey uncertainty.

Table 2: This environmental comparison table is informative but the sources should be more specifically cited. The current citation format (references 10, 11, 13, 15) does not clearly indicate which data points come from which sources. Some values appear to be estimates or ranges, and this uncertainty should be acknowledged.

Figure 2 legend: The legend uses "Metallic" while the text consistently uses "metal." Terminology should be consistent throughout.

4. Writing and Terminology

Line 33-34: The phrase "Patient discomfort remains a primary barrier from maintaining consistent patient care" is grammatically awkward. Consider revising to "Patient discomfort remains a primary barrier to maintaining consistent patient care."

Speculum/specula: The manuscript uses "speculums" throughout. While both "speculums" and "specula" are acceptable plural forms, the authors should verify PLOS ONE style preferences for consistency.

5. Discussion and Interpretation

Generalizability (Lines 164-167), the authors appropriately note limited demographic diversity, but additional demographic information would help readers assess generalizability. The manuscript mentions participants identified as "cisgender women" but does not report the full demographic breakdown (race/ethnicity, socioeconomic status, rural/urban residence). Given that West Virginia has unique demographic characteristics, this context would be valuable.

Environmental concern interpretation (Lines 203-208), the interpretation that younger respondents' comfort outweighs environmental concerns is reasonable but somewhat speculative. The survey design does not appear to have directly assessed this trade-off. The authors should temper this conclusion or clarify if direct trade-off questions were asked.

6. Data Availability

The data availability statement indicates "The data is and will continue to be stored in Qualtrics. It can be made available upon request." PLOS ONE generally requires data to be publicly available. The authors should either deposit de-identified data in a public repository or provide detailed justification for restricted access, including specific contact information for data requests.

Minor Issues

Line 47 contains an awkward superscript placement: "experience.1,2,3" should likely have a space before the citation or use consistent formatting.

Reference 8 cites "Verified Market Reports" which is a commercial market research source. The authors should consider whether a peer-reviewed source is available for this market data, or acknowledge the limitations of commercial market reports.

The SRQR checklist is mentioned (Lines 116-117) as being included in Supplement A, but I could not verify its inclusion in the submitted materials. The authors should ensure this supplement is included with the final submission.

In Summary, this manuscript addresses a practical clinical question with relevance to both patient experience and healthcare sustainability. The study is well-conceived and the discussion provides thoughtful clinical context. The identified issues are minor and addressable through revision. With attention to the methodological clarifications, statistical reporting, and presentation issues noted above, this manuscript will make a valuable contribution to the gynecologic literature. I recommend acceptance after minor revisions.

.

Reviewer #1: **Yes:** Faisal AlkulaibFaisal AlkulaibFaisal AlkulaibFaisal Alkulaib

Reviewer #2: **Yes:** Ahmed Abdullah AlshehriAhmed Abdullah AlshehriAhmed Abdullah AlshehriAhmed Abdullah Alshehri

---

## [Author Response · Author response to Decision Letter 1]

12 Mar 2026

Dear Reviewers,

We have addressed all the issues noted in the revision suggestions. I have noted the changes based on the line numbers of the new document rather than the old as many things have shifted lines.

Line 20 shows the corrected discrepancy between the IRB dates and recruitment period start date. We apologize for the confusion.

Line 34-35 (previously lines 33-34) was revised to avoid grammatically award phrasing.

Line 49 (previously line 47) was revised for its inconsistent superscript placement.

Lines 62-66 & Lines 71-89 use the new reference that replaced the old reference 8 that was not a peer reviewed research article. After finding appropriate references, more lines were added in for context as they added more to our paper. In addition, we removed and replaced two other website sources (old references 10&11) that were not peer reviewed articles with an appropriate reference (new reference 8) that also added more to the paper.

Lines 120-123 & Lines 218-219 address the issue of the lack of power calculation and those implications with the potential of being underpowered to detect small subgroup differences.

Lines 125-129 address survey instrument validation, piloting, and whether there were previously validated instruments for addressing this topic.

Lines 132-135 addressed the response rate and how many patients were approached and did not complete the survey. Potential selection bias has been reported in this section.

Lines 147-150 address the lack of formal correction for multiple comparisons following the chi-square results and how the findings should be interpreted as hypothesis-generating.

Lines 174-174 provide the chi-square statistic, degrees of freedom, and P-value.

Lines 152-154-149 & Line 200 below Table 1 address the fact that some percentages may not total to 100% due to rounding.

Lines 163-167 address how incomplete or missing data was handled as well as how many surveys were submitting in full versus partially complete and how many were used for the primary outcome of this paper.

Lines 223-229 address the lack of collection of demographics in patients as well as the predominantly non-Hispanic White population of West Virginia. This source is straight from the U.S. Census Bureau. If it would be preferred for a peer-reviewed article to be used, that can be addressed.

Lines 280-2292 rephrases the speculative statement regarding environmental concerns affecting speculum preferences. Clarification that no trade-off decision making between comfort and sustainability was assessed was added as well.

Regarding deciding between using speculums or specula, we decided to be consistent with common usage in biomedical literature that PLOS style suggests in order to emphasize accessible language rather than strict Latin plurals. Therefore, we retained “speculums” throughout the manuscript for consistency instead of “specula.”

Figure 2 has been revised to replace “metallic” with “metal” to stay consistent with the rest of the paper.

Table 2: In regard to more accurately citing the references used for table 2, we found a more accurately applicable source for this table and cited each row by source. Therefore, we edited the table and added in more to the discussion centered around this new reference. The added discussion is seen in lines 295-301.

Lines 339-241 were added to address the data availability. It has been made publicly available via the link provided.

Overall, some relocation of sentences and grammar editions have been implemented. Additionally, the references have been re-ordered based on where they fall in number within the paper. These track changes have been noted but not pointed out in this document as they are ultimately the same reference.

Thank you for considering our study and suggesting these helpful revisions. If you have issues accessing the data or have other suggestions, please let us know. We would be happy to revise again.

Sincerely,

Torren Kalaskey

---

## [Decision Letter · Decision Letter 1]

24 Mar 2026

Comfort or Conservation? Investigating Patient Choices Between Plastic and Metal Speculums

PONE-D-25-65007R1

Dear Dr. Kalaskey,

We’re pleased to inform you that your manuscript has been judged scientifically suitable for publication and will be formally accepted for publication once it meets all outstanding technical requirements.

Kind regards,

Alessandro Favilli, PhD, MD

Academic Editor

PLOS One

Additional Editor Comments (optional):

Reviewers' comments:

Reviewer's Responses to Questions

**Comments to the Author**

Reviewer #1: (No Response)

Reviewer #2: All comments have been addressed

2. Is the manuscript technically sound, and do the data support the conclusions?

Reviewer #1: Yes

Reviewer #2: Yes

3. Has the statistical analysis been performed appropriately and rigorously?

Reviewer #1: Yes

Reviewer #2: Yes

4. Have the authors made all data underlying the findings in their manuscript fully available?

Reviewer #1: Yes

Reviewer #2: Yes

5. Is the manuscript presented in an intelligible fashion and written in standard English?

Reviewer #1: Yes

Reviewer #2: (No Response)

Reviewer #1: The following minor points could be considered to further enhance the discussion:

1- The discussion of limitations could be slightly expanded to include the potential for social desirability bias, particularly concerning environmental awareness. Participants may have over-reported their concern for the environment as it is a socially desirable attitude, which might not reflect their true influence on their preferences in a clinical setting.

2- For future research, they might consider suggesting a study design that could more directly explore this, such as a discrete choice experiment. This would be a valuable next step to build upon their current exploratory findings.

Reviewer #2: (No Response)

.

Reviewer #1: **Yes:** Faisal AlkulaibFaisal AlkulaibFaisal AlkulaibFaisal Alkulaib

Reviewer #2: No

---

## [Editor Report · Acceptance letter]

PONE-D-25-65007R1

PLOS One

Dear Dr. Kalaskey,

I'm pleased to inform you that your manuscript has been deemed suitable for publication in PLOS One. Congratulations! Your manuscript is now being handed over to our production team.

Kind regards,

on behalf of

Dr. Alessandro Favilli

Academic Editor

PLOS One